# Internet Addiction as a Moderator of the Relationship between Cyberhate Severity and Decisional Forgiveness

**DOI:** 10.3390/ijerph19105844

**Published:** 2022-05-11

**Authors:** Justyna Mróz, Kinga Kaleta

**Affiliations:** Jan Kochanowski University of Kielce, Department of Psychology, 25-029 Kielce, Poland; kinga.kaleta@ujk.edu.pl

**Keywords:** internet addiction, cyberhate, forgiveness

## Abstract

(1) Background: Cyberhate is becoming increasingly prevalent, just as Internet addiction. One way to deal with hate speech may be to make a decision to forgive the offence. However, addiction to the Internet, due to cognitive changes caused, can play a role in the making of this decision. (2) Methods: A total of N = 246 participants completed the Online Cognitive Scale (OCS), Decision to Forgive Scale (DTFS), and a single-item scale to assess cyberhate severity. In our cross-sectional study, we tested the moderating role of Internet addiction in the relationship between the severity of cyberhate and decisional forgiveness. (3) Results: The results of our study show an inverse correlation between cyberhate severity and decisional forgiveness. We found that Internet addiction moderated the relationship between the perceived severity of cyberhate and forgiveness. In case of a high level of Internet addiction, the transgression severity–forgiveness link is not significant. (4) Conclusions: These results are in accordance with the studies that showed the negative effects of Internet addiction on cognitive processes.

## 1. Introduction

### 1.1. Online Hate Speech

Cyber-aggression is a negative consequence of the invention of the World Wide Web, social networking sites in particular. Services such as Twitter, Facebook, and Instagram have become a place where, in addition to acquiring knowledge, one is likely to be exposed to aggressive behavior [1].

There are different forms of cyber-aggression, for example, cyberbullying [2], phubbing [3], and cyberhate [4]. According to the International Network Against Cyber Hate (INACH), cyberhate is defined as hate speech that takes place online, which refers to “intentional or unintentional public discriminatory and/or defamatory statements; intentional incitement to hatred and/or violence and/or segregation based on a person’s or a group’s real or perceived race, ethnicity, language, nationality, skin colour, religious beliefs or lack thereof, gender, gender identity, sex, sexual orientation, political beliefs, social status, property, birth, age, mental health, disability, disease” [5]. Haters use any textual, video, or photographic means of communication available on the Internet with the aim of discriminating, humiliating, and marginalizing other people [4]. Social networking sites, dating sites, blogs, online games, e-mail, and messengers are places where cyberhate can appear.

Between 20 January and 29 February 2020, the OpCode partners and INACH have conducted a series of shadow monitoring exercises of illegal hate speech on social media platforms in five European countries: Romania, Estonia, Slovakia, Poland, and Spain. The report showed that the most frequent types of reported hate speech are targeting members of sexual minorities (Poland, Estonia), the Roma community (Romania, Slovakia), refugees (Slovakia, Estonia), and that hate speech is antisemitic (Poland) and xenophobic (Spain). In addition to the above-mentioned major grounds of reported illegal content in each country, the report pointed to other groups also exposed to hate speech. Cyberhate is a common experience of Internet users, and it is a form of transgression toward other people in the online world. Wachs et al. [6] showed that being a victim of cyberhate was associated with many variables, including contact with unknown people online, witnessing cyberhate, and excessive Internet use.

Cyberhate is a difficult and stressful situation. The ways to cope with cyberhate include seeking close support, ignoring it, assertiveness, blocking the online-hate offender, and reporting it to the social networking website [7,8]. Another form of dealing with interpersonal conflicts is forgiveness [9,10]. Given the lack of work considering forgiveness as one form of coping with cyberhate, the research presented here fills this gap. In our study, we attempted to determine the relationship between the subjective transgression severity and forgiveness. In addition, the moderating role of Internet addiction was taken into account with reference to previous studies [6].

### 1.2. Transgression Severity and Forgiveness

Transgression severity refers to the level of the negative affect victims experience following a relational offence [11]. Most often, the severity of the wrongdoing is considered in the context of forgiveness. Transgression severity is acknowledging a key predictor of forgiveness [12]. The more severe the wrongdoing, the less likely one is to forgive [13]. The first step in the forgiveness process is making a decision not to seek revenge or avoid the wrongdoer and to control one’s own behavior toward them. Decisional forgiveness may begin with the change of emotions, behavior, and attitude toward the transgressor [14]. The study will assess whether making a decision to forgive may depend on the severity of the transgression.

Forgiveness is conceptualized as a coping strategy to reduce the stressful reaction to a transgression [10]. When the individuals forgive, they recognize the wrongdoing has occurred. They change their own thoughts, emotions, and behaviors so that their responses to it are no longer negative. Forgiveness has positive consequences for both relationships with others and with oneself. Through forgiveness, relationships with others can continue to be friendly and grow. In turn, the positive changes that occur after forgiveness promote better mental and physical health [14].

Relationships between transgression severity and forgiveness have been confirmed by many studies [12,13]. Schulz, Tallman, and Altmaier [15] indicated that there is a link between the severity of a distressing event and the levels of revenge and avoidance among people who experienced a significant interpersonal transgression. Although transgression severity has been linked to forgiveness, no study so far has investigated in what manner Internet addiction might affect this association, in particular, with reference to cyberhate. The link between transgression severity and forgiveness is well-documented [12,13]. The next step is therefore to explore the potential factors that may be relevant to this relation.

### 1.3. Internet Addiction

With the invention of the World Wide Web, problematic Internet use and addiction from various forms of Internet-related activity, for example, social media addiction [16], Facebook addiction [17], Instagram addiction [18], and online game addiction [19], have become a growing problem.

Among many psychological models and theories concerning problematic Internet use or Internet addiction [20], the cognitive-behavioral model of problematic Internet use deserves considerable attention [21]. According to this model, problematic Internet use is a stress coping mechanism, and it refers to behavior focused on a compulsion to be online and communicate with others [21]. Moreover, Fontana et al. [22] indicated problematic Internet use as maladaptive coping strategy. Individuals with a high level of problematic Internet use withdrew aggressive emotions toward people who used hate toward them. Studies based on Davis’s model indicated a negative link between Internet addiction and mental health [23] and self-compassion [24]. Moreover, a few studies showed the negative effects of Internet addiction on cognitive processes [25,26]. Individuals addicted to the Internet have impaired cognitive flexibility [27], problems dealing with emerging cognitive conflicts [26], impaired inhibitory control, and diminished cognitive efficiency [28]. According to previous studies, the difficulties with the regulation and organization of one’s own behavior and flexible adaptation may make it difficult to properly assess the experienced events, e.g., transgressions, and to make appropriate decisions and present proper behavior. Notably, the previous studies showed a positive correlation between excessive Internet use and cyberbullying victimization [29] and excessive Internet use and experiencing cyberhate [6]. The individuals who spent much time online are more exposed to cyberhate. Internet addiction and forgiveness were examined in only a few studies [30,31]. Arslan [30] showed the relationship between psychological maltreatment and Internet addiction, which was moderated by the tendency to forgive. On the other hand, Wang and Qi [31] indicated that a high level of forgiveness inhibits the indirect effect of harsh parenting on problematic Internet use.

### 1.4. Aim of the Study

The aim of the present study is to examine the relationships between cyberhate severity, Internet addiction, and decisional forgiveness. The potential moderating effect of Internet addiction on the relationship between cyberhate severity and decisional forgiveness was also explored. We hypothesized that Internet addiction would distort (moderate) the relationship between transgression severity and decisional forgiveness.

## 2. Materials and Methods

### 2.1. Participants and Data Collection Procedures

We used a Polish sample of 246 participants. The respondents were requested to participate in the study voluntarily—no remuneration was offered to them. They were given paper-and-pencil questionnaires to answer all the questions in private and then asked to return the completed questionnaires. The female participants accounted for 72.5% (*n* = 171) of the sample and male participants for the remaining 27.5% (*n* = 64). The subjects’ ages ranged from 18 to 50 years, with a mean of 23.72 (*SD* = 6.97). They usually lived in cities (53.6%), then in the country (37.32%), and less often in towns (12.8%). They were residents of cities (53.6%), rural areas (37.2%), and less frequently, residents of towns (12.8%). As regards the level of education, 64.7% of the participants had completed secondary education school, whereas 27.2% had higher education once graduated from university and 7.2% had college education from post-secondary school. All the participants met two enrolment criteria: they were adults and reported having experienced cyberhate in the past. Hate speech refers to abusive commentary, systematic spreading of misinformation about a person, chauvinism, ridiculing, providing incorrect information about a person, posting intimate photos on different web sites, threats, and blackmail.

### 2.2. Measures

#### 2.2.1. Cyberhate Severity

Perceived cyberhate severity was measured using only one item. The participants were asked to indicate severity of the transgression. They used a five-point rating scale from 1 (suffering to a small degree) to 5 (suffering to a great extent).

#### 2.2.2. Decisional Forgiveness

Decisional forgiveness was measured with the Decision to Forgive Scale [32]. We used the Polish adaptation of this scale [33]. The scale consists of 5 items participants have to rate using a 5-point scale, from 1 (strongly disagree) to 5 (strongly agree). Scores range from 5 to 25, with higher scores indicating a stronger decision to forgive one specific offense. Sample item: “My choice is to forgive him or her”, “I made a commitment to forgive him or her”. Cronbach’s alpha is 0.90.

#### 2.2.3. Internet Addiction

Internet addiction was measured using the Polish version of the Online Cognition Scale (OCS). This instrument contains 36 items and uses a seven-point Likert-type scale (1—strongly disagree to 7—strongly agree). It was developed by Davis, Flett, and Besser [21] to assess Internet addiction, and it has four sub-dimensions: loneliness/depression (6 items), diminished impulse control (10 items), distraction (7 items), and social comfort (13 items). A sum of all scores yields a total score ranging from 36 to 252, with higher scores indicating higher Internet addiction level. Sample items: “My use of the Internet sometimes seems beyond my control”, “When I have nothing better to do, I go online”. The internal consistency coefficient of the Polish version is 0.80 and the test–retest reliability coefficient is 0.87. Polish adaptation of this scale was prepared by Błachnio, Przepiórka, and Hawi [34].

### 2.3. Statistical Analysis

The moderation model was tested separately for general IA and four dimensions of IA (loneliness/depression, diminished impulse control, distraction, and social comfort) as moderators using the PROCESS 3.3. macro for SPSS ver. 21. A bootstrapping approach was used to examine the indirect effect on each of the 5000 bootstrapped samples from the original dataset using random sampling with replacement. Statistical significance was measured using percentile-based confidence intervals (Cis) around these estimates. A bootstrap confidence interval (95% CI) which does not include the “0” value signals a significant effect.

## 3. Results

Table 1 presents correlation coefficients between cyberhate severity, Internet ad-diction, and dispositional forgiveness. Cyberhate severity showed a negative correlation with decisional forgiveness. We found no correlation between Internet addiction and the severity of cyberhate and decisional forgiveness.

To test for moderation, we used hierarchical linear regression [35].

### Internet Addiction as a Moderator

We conducted linear regression to investigate the effect of cyberhate severity and Internet addiction on decisional forgiveness. We performed a moderation analysis with one moderator in five models, each model exploring a specific subscale of Internet addiction: loneliness/depression, diminished impulse control, distraction, and social comfort. Transgression severity was an independent variable and decisional forgiveness was a dependent variable; we used the conceptual Model 1 according to Hayes [36]. The results (Table 2) showed that in the case of a low or medium level of Internet addiction, the relationship between cyberhate severity and decisional forgiveness was significant. The relation between cyberhate severity and decisional forgiveness was negative. However, when IA was high, this relationship was not significant (Figure 1).

The same procedure was applied to the subscales of Internet addiction, social comfort, loneliness, diminished impulse control, and distraction, used as moderators. Just like a general Internet addiction, low and medium levels of loneliness, diminished impulse control, and distraction moderated the relationship between the severity of cyberhate and decisional forgiveness. On the other hand, high subscale scores revealed no links between the severity of cyberhate and forgiveness. Only social comfort failed to moderate the relationship between cyberhate severity and decisional forgiveness in any way.

## 4. Discussion

The study examined the relationship between cyberhate severity, Internet addiction, and decisional forgiveness. We specifically tested the hypotheses concerning the moderating role of Internet addiction in the relationship between cyberhate severity and decisional forgiveness, and we determined how Internet addiction moderated the relationship between the perceived severity of a transgression and decisional forgiveness. We postulated that Internet addiction may disturb the apparent relationship between cyberhate severity and decisional forgiveness.

Our results confirmed the relationship between cyberhate severity and decisional forgiveness. According to scholars, there is an unquestionable link between the perceived severity of a transgression and forgiveness [12,37]. If a transgression is perceived as relatively minor, it is more likely to be forgiven. Deeply hurt people, on the other hand, find it harder to forgive [38]. Forgiveness requires a greater effort from them, e.g., Waldron and Kelly [38] indicated that direct strategies (e.g., discussing and being explicit about the transgression) are likely to induce forgiveness when the transgression is perceived as more severe. Our study makes a contribution to the knowledge concerning the link between wrongdoing severity and forgiveness by confirming the moderating role of Internet addiction. To our knowledge, Internet addiction, transgression severity, and forgiveness have not been tested in the same model, especially as far as cyberhate experience is concerned.

The results partially supported our hypotheses concerning the moderating role of Internet addiction. It showed that Internet addiction moderates the relationship between cyberhate severity and decisional forgiveness in that, in individuals with scores representing low or medium levels of Internet addiction, a more strongly perceived cyber-transgression severity was related to lower decisional forgiveness. Along with greater Internet addiction, the relationship between cyberhate severity and decisional forgiveness was becoming less significant. A similar dependence was reported for three subscales describing problematic Internet use: loneliness/depression, diminished impulse control, and distraction. These findings are in line with the previous studies which showed that problematic Internet use was moderated by the link between aggression and internalizing problems when its level was low or medium. When problematic Internet use was high, this link was nonsignificant [22].

The results obtained support the thesis that IA is a maladaptive strategy for coping with difficult situations. Fontana et al. [22] interpreted problematic Internet use as a strategy for helping individuals to contain negative emotions toward others. Thus, IA as a maladaptive coping strategy supports the control of negative feelings when cyberhate is experienced. Cyberhate is not perceived as an experience of wrongdoing, and there is no need to forgive the other person.

The lack of a link between cyberhate severity and forgiveness when the level of IA was high can also be interpreted by reference to the results of previous studies. Among the negative effects of Internet use, they indicated cognitive distortions and the development of a defense mechanism [25]. The functioning of cognitive processes is important in diagnosing Internet addiction. Cognitive control facilitating flexible adaptation to environmental requirements may be distorted in people with an Internet addiction [25], which may be manifested in impaired executive functions. For example, Dong, Zhou, and Zhao [26] suggested that Internet addicts have an impaired ability to resolve cognitive conflicts. Moreover, individuals with more severe symptoms of Internet addiction, owing to an impaired ability to update and monitor the incoming information, are likely to be blind to problematic situations. They have difficulty with interpreting information appropriately. Therefore, when they experience a cyber-transgression, they either fail to interpret it as harmful behavior, or they ignore it because they have no skills to deal with it due to difficulty in resolving conflicts. Additionally, in electrophysiological studies, Jiao et al. [39] showed that people with symptoms of Internet addiction have reduced affective arousal and the ability to share other people’s pain. This may also suggest that their own emotional pain due to a transgression will be reduced as a result of lower affective arousal. Therefore, the apparent transgression severity–forgiveness link became insignificant as cyberhate victims addicted to the Internet overlook their own wrongdoings.

The study is not free from limitations. First of all, it was a cross-sectional study; therefore, it was not possible to determine the causality. A longitudinal study would provide more precise findings related to the link between Internet addiction, transgression severity, and forgiveness. The next limitation was the use of self-report measures. Moreover, an important limitation was the method of assessing the transgression. First, we were unable to distinguish between single and repeated transgressions, which may be related to the decision to forgive or problematic Internet use. Second, we only used the perceived severity of a transgression. Fincham et al. [12] pointed to differences between an objective and subjective assessment of transgression severity. Additionally, we measured the perceived severity of transgressions using only one item.

The sample used in the study was relatively small and dominated by young people and females, so we failed to control gender and age. To improve the generalizability of the results, future studies must include more diversified populations and individuals who experienced specific types of cyber-transgressions, such as cyberbullying, hate speech, phubbing, etc.

## 5. Practical Implication

The results of the current study speak to the broader issue of people who use the Internet excessively. Considering the current outcomes, it might be advisable to include Internet use measurements in routine psychiatric assessments. These results also show that interventions for the prevention of IA should focus on enhancing the quality of cognitive skills and emotional regulation in order to reduce the psychiatric symptoms [25]. Improved cognitive functioning will support a more relevant assessment of the situation, particularly the assessment of harm and decision making relating to coping, such as forgiveness.

## 6. Conclusions

The relationship between Internet addiction and mental health problems, such as depression [39,40,41], anxiety [41], or insomnia [39], may help account for the lack of a link between perceived transgression severity and decisional forgiveness when symptoms of IA are the most severe. Depression and anxiety disorders pointed to impairment within the cognitive process and executive functions, e.g., [42].

## Figures and Tables

**Figure 1 ijerph-19-05844-f001:**
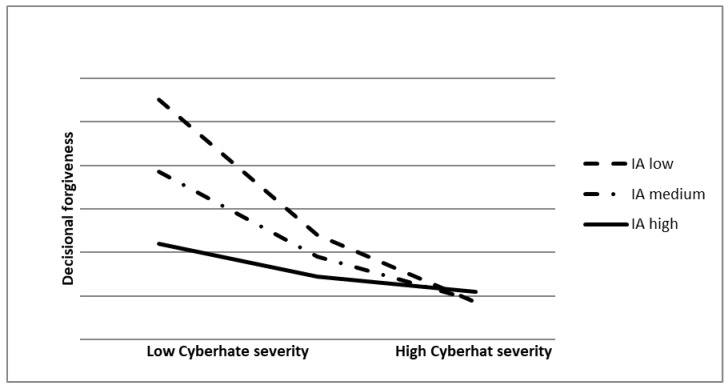
Graphical depiction of moderation results. Note: IA—Internet addiction.

**Table 1 ijerph-19-05844-t001:** Intercorrelations (Pearson’s r) between analyzed variables.

	1	2	3	4	5	6	7
Cyber-transgression severity	-						
Internet addiction	−0.009	-					
Social comfort	−0.022	−0.845 *	-				
Loneliness/depression	−0.045	0.865 *	0.658 *	-			
Diminished impulse control	0.045	0.874 *	0.643 *	0.736 *	-		
Distraction	−0.002	0.701 *	0.361 *	0.522 *	0.506 *	-	
Decisional forgiveness	−0.232 *	−0.087	−0.115	−0.052	−0.115	0.010	-

Note: * *p* < 0.01.

**Table 2 ijerph-19-05844-t002:** Results of moderation analysis for the independent variable—cyberhate severity—and dependent variable—decisional forgiveness.

Moderator	R^2^ch	B	t	*p*	95% CI	Interaction					
						B_L_	*p* _L_	B_M_	*p* _M_	B_H_	*p* _H_
Internet addiction	0.025	0.02	2.46	0.014	[0.042; 0.382]	−1.73	0.000	−1.06	0.000	−0.42	0.24
Social comfort	0.006	0.02	1.27	0.203	[−0.015; 0.071]						
Loneliness/depression	0.022	0.09	2.33	0.020	[0.014; 0.172]	−1.78	0.00	−1.40	0.00	−0.38	0.32
Diminished impulse control	0.027	0.07	2.57	0.010	[0.016; 0.124]	−1.66	0.00	−1.10	0.00	−0.25	0.52
Distraction	0.027	0.09	2.64	0.009	[0.022; 0.152]	−1.94	0.00	−0.98	0.00	−0.28	0.47

Note: _L_—value of low moderator; _M_—value of medium moderator; _H_—value of high moderator.

## Data Availability

The dataset presented in this study is available on reasonable request from the corresponding author.

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
