# Peer review of "Internet Addiction as a Moderator of the Relationship between Cyberhate Severity and Decisional Forgiveness"

_ijerph, 2022, doi:10.3390/ijerph19105844_

Round 1

Reviewer 1 Report

Internet addiction as a moderator of the relationship between cyberhate severity and decisional forgiveness

This is an interesting article with important implications for the field of internet addiction, cyberhate and decisional forgiveness. Nevertheless, I believe the following changes would improve the overall quality of the manuscript:

  1. The paper would benefit from a deeper literature review, especially regarding psychosocial consequences of the three main constructs.
  2. Please provide more sociodemographic information of the participants.
  3. Also, please provide more information regarding the sampling methods, inclusion criteria, etc.
  4. Please provide examples of items.
  5. Please clarify if instruments used were Polish versions or how were they adapted and validated for Poland.
  6. Please include an implications section. Discuss how these results will be helpful when dealing with addiction, from a public health perspective, and from clinical perspective.

Best wishes.

Author Response

Respected Reviewer 1,

Thank you very much for your review. We appreciate your suggestions and comments,

which will undoubtedly improve the quality of our manuscript.

Internet addiction as a moderator of the relationship between cyberhate severity and decisional forgiveness

Point 1. The paper would benefit from a deeper literature review, especially regarding psychosocial consequences of the three main constructs.

Answer: We have completed the introduction in several places so that the information is more complete.

Point 2. Please provide more sociodemographic information of the participants. Also, please provide more information regarding the sampling methods, inclusion criteria, etc.

Answer:The information was completed. (lines 124-133).

Point 3. Please provide examples of items.

Answer: Exampels of items were given earlier, formatted to make them more visible.

Point 4. Please clarify if instruments used were Polish versions or how were they adapted and validated for Poland.

Answer:  As stated in the text, we used Polish versions DTFS and OCS

Point 5. Please include an implications section. Discuss how these results will be helpful when dealing with addiction, from a public health perspective, and from clinical perspective.

Ansewer: We inlcluded practical implications section.

Reviewer 2 Report

The paper “Internet addiction as a moderator of the relationship between cyberhate severity and decisional forgiveness” addresses an interesting problem. However, the current version of the paper shows several limitations either from a theoretical and a methodological point of view.
Introduction: The first part of the introduction underlines the topic of the problem. However, the second part of the literature review is vague and the relationship among the variables of the study is not discussed in a proper way. For example, the authors should justify from a theoretical point of the view the role of the variables and in particular the effect of the moderator in the proposed association. There are a lot of published studies on cyberhate, which should be discussed in this paper.

Method 
Cyberhate severity. There are some measures that try to assess Cyberhate. The reason why the authors used only one item should be explained. I suggest including the item.

Results
There is no correlation between Cyberhate severity and Internet addiction (Table 1)! I think that the authors should reconsider the relations between the variables of the study.

Internet addiction as a moderator: Please, check the statement (lines 153-156, p. 4). The text is different compared to the result in Figure 1.

Table 2. Please report in a correct format the results of the moderation analysis. Additionally, please check how to report the CI. In general, Table 2 is hard to read.

Discussion
The content of the discussion should be revised according to the results of the study. 

Author Response

Respected Reviewer 2,

Thank you very much for your review. We appreciate your suggestions and comments, which will undoubtedly improve the quality of our manuscript

 Point 1. Introduction: The first part of the introduction underlines the topic of the problem. However, the second part of the literature review is vague and the relationship among the variables of the study is not discussed in a proper way. For example, the authors should justify from a theoretical point of the view the role of the variables and in particular the effect of the moderator in the proposed association. There are a lot of published studies on cyberhate, which should be discussed in this paper.

Answer: Thank you very much for your comment .We have completed the introduction. For example underlines the topic of the problem (lines 48-58, page 1).

Point 2. Method

Cyberhate severity. There are some measures that try to assess Cyberhate. The reason why the authors used only one item should be explained. I suggest including the item.

Answer: Thank you very much for your suggestion.We have completed this information.

Point 3. Results

There is no correlation between Cyberhate severity and Internet addiction (Table 1)! I think that the authors should reconsider the relations between the variables of the study.

Answer: To our knowledge, there is no need for a moderator variable to be correlated with either of the others variables, especially with the independent one (f.e. Hayes, Rockwood, 2017).

Point 4. Internet addiction as a moderator: Please, check the statement (lines 153-156, p. 4). The text is different compared to the result in Figure 1.

Answer: The sentence was described in correct way.

Point 5. Table 2. Please report in a correct format the results of the moderation analysis. Additionally, please check how to report the CI. In general, Table 2 is hard to read.

Answer: Table was corrected. We added the Note and corrected CI.

Point 6. Discussion

The content of the discussion should be revised according to the results of the study.

Answer:We have completed discussion (lines 240-249 page 6-7).

Round 2

Reviewer 1 Report

Thank you for implementing all the requested changes. They have improved the overall quality of the article.

Best wishes.